

# Prevalence of non-specific chronic low-back pain and risk factors among male soldiers in Saudi Arabia

Mohammad Sidiq[1,2], Wadha Alenazi[1], Faizan Z. Kashoo[3], Mohammad Qasim[1], Marisia Paz Lopez[1], Mehrunnisha Ahmad[4], Suresh Mani[5], Mohammad Abu Shaphe[6], Omaymah Khodairi[7], Abdulqader Almutairi[7] and Shabir Ahmad Mir[8]

[1] Department of Rehabilitation Medicine, Northern Area Armed Force Hospital, Hafer Al-Batin, Hafer, Saudi Arabia
[2] Faculty of Physiotherapy, Madhav University, Abu Road, Rajasthan, India
[3] Department of Physical Therapy and Health Rehabilitation, College of Applied Medical Sciences, Mjamaah University, Al Majmaah, Saudi Arabia
[4] Department of Nursing, College of Applied Medical Sciences, Majmaah University, Almajmaah, Saudi Arabia
[5] Department of Physiotherapy, Lovely Professional University, Phagwara, Punjab, India
[6] College of Applied Medical Sciences, Physical Therapy Department, Jazan University, Jazan, Saudi Arabia
[7] Department of Physical Therapy, College of Applied Medical science, Jazan University, Jazan, Saudi Arabia
[8] Department of Medical Laboratory Sciences, College of Applied Medical Sciences, Majmaah University, Majmaah, Riyadh, Saudi Arabia

Corresponding author
Faizan Z. Kashoo,
f.kashoo@mu.edu.sa

## ABSTRACT

**Background.** Non-specific chronic low back pain (NSCLBP) is the most common musculoskeletal disorder affecting health and work among the military population. NSCLBP is a complex disorder with several risk factors contributing to its occurrence. Therefore, the objective of our study was to estimate the prevalence and contribution of risk factors towards NSCLBP among male soldiers in Saudi Arabia.

**Methods.** A cross-sectional study was conducted from March 2020 to January 2021 among the military personnel at Hafar Al-Batin military base, Saudi Arabia. The entire population ($n = 62,000$) at the military base were invited to participate in the study. The participants were invited to participate in the study either through direct referral from in-patient and out-patient departments of the military hospital or by invitation through pamphlets, email, and advertisement across the offices and residential areas. Soldiers reporting lower back pain for at least 12 weeks were screened for inclusion criteria at the physical therapy department of the military hospital. Inclusion criteria included pain or discomfort originating from the lower back without any known spinal diseases. Participants with a systemic inflammatory disorder, trauma, neurological symptoms, and recent spinal surgery were excluded. All eligible participants were assessed for demographic variables and risk factors and complete the Rolland Morris Disability Questionnaire and WHO-Five Well-Being Index.

**Results.** This study identified a 46.3% prevalence of pain originating from the spine with a 2.7% prevalence of NSCLBP. Spearman's rho correlation between the severity of disability due to NSCLBP was strongly associated with age ($r_s = 0.834$, $p < 0.01$), quality of sleep ($r_s = 0.790$, $p < 0.01$), body mass index (BMI) ($r_s = 0.617$, $p < 0.01$), smoking ($r_s = 0.520$, $p < 0.01$), co-morbidity ($r_s = 0.357$, $p < 0.01$), but not with the level of physical activity ($r_s = 0.044$, $p = 0.07$).

**Conclusion**. There was a high prevalence of pain originating from the spine among male Saudi soldiers with a relatively low prevalence of NSCLBP. However, the prevalence of disability due to NSCLBP was strongly associated with age, sleep quality, BMI, smoking habit, and co-morbidity.

# INTRODUCTION

Low back pain (LBP) is one of the most recurring medical complaints that require health care intervention. It is the most frequent type of musculoskeletal disorder causing significant disability and job absenteeism (*Alnaami et al., 2019*; *Cunningham, Flynn & Blake, 2006*). More than half of the general population seeks medical advice for LBP at some point in their lives (*Ferreira et al., 2010*). The global prevalence of LBP among the general population ranges between 15% to 45% (*Al-Arfaj et al., 2003*) and incurs a high medical cost. LBP is a multi-factorial condition, and the evidence does not always support a clear cause and effect relationship (*Hartvigsen et al., 2018*). Back pain without any known cause for 12-weeks is termed non-specific chronic low back pain (NSCLBP) (*Rozenberg, 2008*). However, there are known risk factors associated with NSCLBP. These factors can be categorized into individual and environmental factors. Individual factors include a sedentary level of activity, age, smoking status, obesity, psychological stress, history of back pain, and pre-existing conditions (*Hulla et al., 2019*). The environmental factors include lifting technique, physical activity required at work, standing time, and sitting duration at work (*Burström, Nilsson & Wahlström, 2015*; *Campbell, Wynne-Jones & Dunn, 2011*; *Coenen et al., 2014*; *Dario & Ferreira, 2015*; *Lardon et al., 2014*).

Military service begins with an introductory training period. The unique characteristics of military training include both the intensity and duration. Since one of the main goals of the training is to improve physical performance level, the volume of physical activity increases linearly during military training. In Saudi Arabia, the problem of LBP is not distinct from that in other parts of the globe, and the prevalence of LBP among the general population is reported to be 18.8% (*Aldera et al., 2020*). Specific spinal postures and physical activities have been associated with LBP (*Swain et al., 2020*; *Johanning, 2000*). Soldiers are mainly at risk of developing LBP due to their high physical demands of their profession. Active military personnel usually carry heavy equipment during training which can lead to over-stressing of their musculoskeletal system, especially the spine (*Parreira et al., 2018*).

Several risk factors have been reported to be associated with non-specific low back pain among the general public (*Sribastav et al., 2018*). Research is scarce about the risk factors associated with NSCLBP. Only one case-control study was conducted in Iran among 92 military personnel to explore the association between risk factors and the incidence of NSCLBP. The authors of the study reported exercises, smoking, body mass index (BMI),

education level and bad posture are the potential risk factors associated with NSCLBP (*Ramezani et al., 2015*). Therefore, identifying the other additional risk factors such as quality of sleep, age, co-morbidity towards the occurrence of NSCLBP in the military population may give us a better insight into its prevention strategies and treatment.

NSCLBP can be effectively managed with patient education, postural correction, and manual therapy (*Stochkendahl et al., 2018*). Conversely, untreated NSCLBP may lead to poor prognosis and disability (*Nieminen et al., 2021*). The medical cost involved in the management of NSCLBP is relatively lower than the more severe form of LBP such as, radiculopathies and intervertebral disc prolapsed (*Bachmann et al., 2009*). Therefore, identifying and treating NSCLBP as soon as possible to onset would significantly reduce medical costs and absenteeism from work (*Miyamoto et al., 2019*).

There are regional studies that have investigated the prevalence of LBP across Saudi Arabia among health care workers. For example, in Riyadh province of Saudi Arabia, 83.9% of medical practitioners suffered from LBP (*Almalki et al., 2016*). The prevalence of LBP among medical workers in the eastern region of Saudi Arabia was 79% (*Al Bahrani et al., 2015*), and 73.9% of health care workers reported suffering from LBP in the southwest region of Saudi Arabia (*Alnaami et al., 2019*). It was reported that 23.9% of rehabilitation professionals seek medical leave due to LBP (*Abolfotouh et al., 2021*). There is no research literature available about the prevalence of NSCLBP among general public or military soldiers in Saudi Arabia. Military personnel are at a greater risk of developing LBP due to the nature of their occupation. Moreover, dietary habits, genetic make-up, culture, and routine physical activity among Saudi soldiers differ from other military across the globe. Therefore, the study aimed to identify the LBP, specifically NSCLBP, and the unique contribution of risk factors towards the occurrence of NSCLBP among Saudi military personnel. The purpose of this cross-sectional study was to estimate the prevalence and risk factors for NSCLBP among soldiers in the Saudi armed forces.

## MATERIALS & METHODS
### Study design, participants and research setting
This cross-sectional study was conducted among the entire military population ($n = 62,000$) at Hafar Al-Baten, Saudi Arabia, from March 2020 to January 2021. The military base comprises administrative buildings, sports complexes, training grounds, residential areas, and general hospitals.

King Fahd Medical City (KFMC) Institutional Review Board approved the study (IRB 19-033E). Hafar Al-baten military base only recruits male soldiers; therefore only male soldiers were enrolled in this study using a purposive sampling method. Participants were invited to participate in the study through pamphlets, emails, and advertisements across the offices and residential areas or referred to physical therapy department from in-patient or out-patient departments of the military hospital. All participants who were under military training or in-service were eligible to participate in this study. Soldiers reporting with LBP were screened for inclusion criteria. Inclusion criteria included pain or discomfort originating from the lumbosacral region without any known cause, which lasted

for at least 12 weeks, without any radiating or specific spinal diseases. The persistence of NSCLBP for 12 weeks is considered chronic (*Hallegraeff et al., 2012*). Participants who were currently suffering from NSCLBP were eligible to participate in the study. The informed consent form was signed by each participant, with the complete information provided about the study procedures. Four physical therapists at the physical therapy department at King Khalid military hospital (Hafar Batin) carried out physical examinations based on a recommended clinical guideline to diagnose NSCLBP (*Oliveira et al., 2018*). A review of medical records, identification of red flags, and radiographic imaging were conducted to identify the systemic cause of LBP. The physical examination was divided into seven sections that included observation (posture and gait), a functional activity that reproduced symptoms, active movement of the lumbar spine and hip, palpation of the lumbar spine and sacral region, assessment of core muscle such as transverse abdominis, multifidus, and pelvic control, a neurological examination which includes myotome, dermatome, sensation, straight leg raise test and femoral nerve test and last was pain related work issues and conflicts (*McCarthy et al., 2006*). Participants with a prolapsed inter-vertebral disk, back surgery, or vertebral column disorders were excluded from the study. However, participants with systemic disorders such as hypertension or diabetes mellitus not related to the cause of backache were included in the study (Fig. 1).

The primary outcome of the study was the prevalence of NSCLBP among military personnel in Saudi Arabia. Assuming a 95% confidence level, with an expected prevalence between 53.2% to 79.17% LBP based upon an earlier study (*Awaji, 2016*) with 0.05 absolute precision, the required sample size was between 255 to 382 military personnel. However, the questionnaire was distributed to the military population at the military base comprising 620000 military personnel. A total of 28706 responded with back pain with a 46.3% response rate.

## Instrumentation

Participants who met the inclusion criteria and signed the consent form were eligible to participate in the study, with the complete information provided about the study procedures. Participants were examined for demographic data such as age, weight, height, years of service and interviewed for role in the military and history of smoking, and medical records were reviewed for any pre-existing conditions. BMI was classified according to the WHO classification as underweight (less than 18.5 kg/m2), normal (18.5–24.9 kg/m2), overweight (25.0–29.9 kg/m2), and obese (30.0 kg/m2 and above) (*Purnell, 2018*). Participants were also categorized into age groups as 20–30, 31–40, and 41–50 years. Each participant had to self-report the Arabic version of Rolland Morris Disability Questionnaire (RMDQ), the Arabic version of Pittsburgh quality of sleep, WHO-Five Well-being Index (WHO-5), and level of physical activity. The severity of disability due to back pain among participants was evaluated by a cross-culturally translated and validated Arabic version of RMDQ (*Maki et al., 2014*). Based on the RMDQ scoring, participants were characterized as mild, moderate, or severe disabilities. The Arabic version of the RMDQ has a high internal consistency (Cronbach α = 0.72) and reliability (ICC = 0.900;

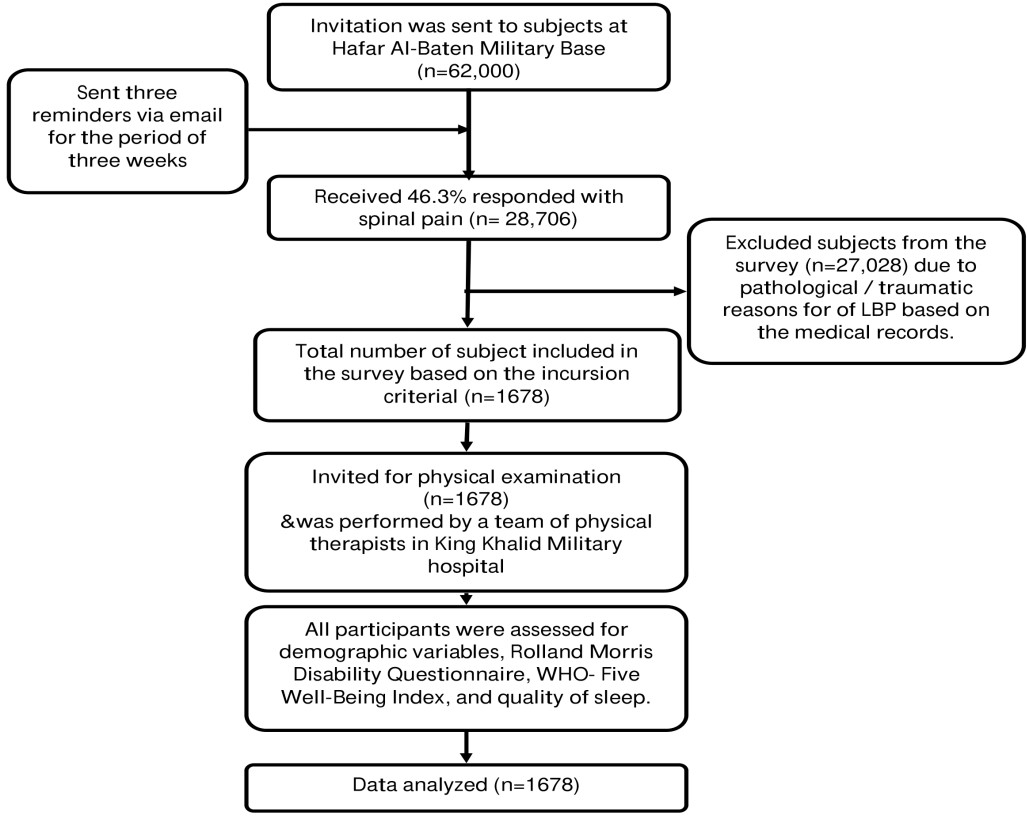

**Figure 1 Procedure of Invitation, recruitment and physical examination of the military personnel.**

95% CI [0.753–0.951]) and good agreement (ICC = 0.925, CI = 0.81–0.97) with the English version of RMDQ (*Maki et al., 2014*).

The quality of sleep among participants was evaluated by an Arabic validated version of PSQI (*Suleiman et al., 2010*). In scoring the PSQI, seven component scores are derived, each scored 0 (no difficulty) to 3 (severe difficulty). The component scores are summed to produce a global score (range 0 to 21). A global score of 5 or more indicates poor sleep quality; the higher the score, the worse the quality. The Arabic version of PSQI is reported to have good reliability, (Cronbach α = 0.77) acceptable internal consistency, (Cronbach α = 0.65) and moderate to a high correlation between global PSQI and five components of PSQI scores (*Al Maqbali et al., 2020*; *Suleiman et al., 2010*).

Quality of life was evaluated by the Arabic version of the WHO-5 (*Sibai et al., 2009*). Participants reported their quality of life status by marking on a 5-point rating scale for five statements. The score was multiplied by 4 to obtain the 0–100 range. Zero stands for the worst possible quality of life, while 100 stands for the best possible quality of life. The Arabic version of the WHO-5 showed good internal consistency and test-retest reliability (Cronbach α = 0.88/r = 0.73) (*Sibai et al., 2009*)

Physical activity level was categorized into four levels based on the number of military drills performed by the participants per week. Sedentary physical activity meant that the

individual was not involved in military drills; light meant that the individual performed military drills once or twice a week; moderate implied thrice or four times a week, and vigorous implied five to all days of a week.

### Statistical procedures

Data were collected in Microsoft Excel (2007) and analyzed by IBM SPSS (version 20). The ordinal scale data such as level of physical activity, BMI, WHO-5, and RMDQ were analyzed using Spearman rank-order analysis. Ordinal regression analysis was conducted between age, years in service, smoking, military rank, quality of sleep, level of physical activity, BMI, and WHO-5, with RMDQ as the independent variable. Descriptive statistics were performed to generate frequency and percentage for each variable.

## RESULTS

Out of 62,000 contacted for the survey, 28,706 (46.3%) responded with pain originating from the spine and 1,678 (2.7%) with NSCLBP (Fig. 2). The mean age of military personnel with NSCLBP was 32.5 ± 7.41 years and a mean BMI of 25.74 ± 3.10 kg/m2. Out of 1,678 participants, 617 (74.5%) had mild, 679 (40.5%) moderate, or 382 (22.8%) severe intensities of NSCLBP.

The younger age group (20–30 years) reported mild ($n = 617$, 87.3%) to moderate ($n = 90$, 12.7%) back pain disability as compared to the older age group. A total of 228 (13.5%) participants in this study were obese, out of which 93 (40.8%) reported severe disability due to NSCLBP. Participants engaging in a vigorous level of physical activity reported mild back pain disability ($n = 222$, 13.2%) compared to sedentary participants ($n = 206$, 65.2%). The NSCLBP prevalence was highest among corporal ranked military officers ($n = 396$, 23.5%) than others. Further analysis revealed that corporal ranked officers were overweight ($n = 216$, 54.5%) and engaged in a moderate level of activity ($n = 182$, 45.9%). Out of 404 active smokers, the majority (91.8%) reported mild back pain disability (Table 1).

Four hundred and four (48%) of participants reported a co-morbidity such as diabetes ($n = 141$, 8.4%), hypertension ($n = 21$, 1.3%), respiratory conditions ($n = 175$, 10.4%), arthritis ($n = 9$, 0.5%), neck pain ($n = 97$, 5.8%), and shoulder pain ($n = 364$, 21.7%).

There was a strong positive correlation between RMDQ with days lost due to back pain, ($r_s(1678) = .902$, $p = .001$), RMDQ with age ($r_s(1678) = .834$, $p = .001$), RMDQ with years in service ($r_s(1678) = .828$, $p = .001$), RMDQ with BMI ($r_s(1678) = .617$, $p = .001$), RMDQ with cigarette smoking ($r_s(1678) = .520$, $p = .001$), RMDQ with quality of sleep ($r_s(1678) = .566$, $p = .001$), and a strong negative correlation between RMDQ with well-being ($r_s(1678) = -.740$, $p = .001$), and RMDQ with working hours ($r_s(1678) = -.681$, $p = .001$). Details of the other significant correlations between variables are given in Table 2.

An ordinal regression analysis was conducted to predict relationships between dependent (RMDQ scores) and independent variables (BMI, age, smoking status, level of activity). An increase in BMI was associated with an increase in the odds of higher scores in the RMDQ, with an odds ratio of 0.106 (95% CI [0.023–0.189]), Wald $\chi 2$ (1) = 6.270, $p = 0.012$. An

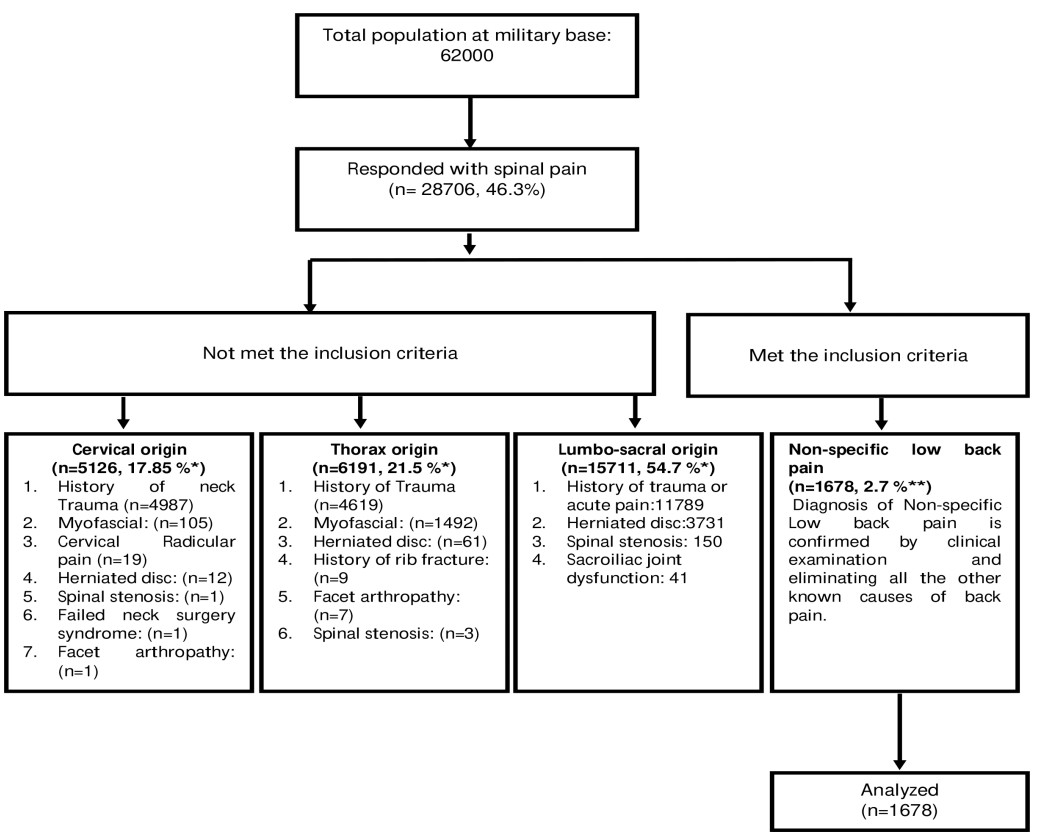

**Figure 2** **Flow chart of the sample population, respondents, inclusion, and analysis.** An asterisk (*) indicates percentage calculated for the participants who responded to have back pain, two asterisks (**) indicates percentage calculated from the sample population.

increase in age (expressed in years) was associated with an increase in the odds of higher scores in the RMDQ, with an odds ratio of 0.494 (95% CI [1.196–1.357]), Wald $\chi 2$ (1) = 95.792, $p < 001$. Participants who did not smoke cigarettes were −1.350 (95% CI [−2.143 to −1.013]) times less likely to score higher on RMDQ than smokers, with a statistically significant effect, Wald $\chi 2$ (1) = 8.964, $p = .003$. Military personal engaging in severe military drills were −1.578 times (95% CI [1.196–1.357]) Wald $\chi 2$ (1) = 6.270, $p = 0.012$, less likely to score higher on the RMDQ than personnel engaging in sedentary and light military drills.

## DISCUSSION

The prevalence of back pain among military personnel was 46.3%, which included pain originating from the lumbosacral region (54.6%), thorax region (21.5%), and cervical region (17.8%). The prevalence of NSCLBP was 2.7% among the population of 62,000 armed forces. The low prevalence of NSCLBP in this study may be due to stringent inclusion criteria. Such as, participants eligible were currently suffering from NSCLBP, pain arising from lumbo-sacral region only and no history of known cause or trauma. A

**Table 1 Demographic variables and grouping of variables for military personal with non-specific chronic low back pain.**

| | | Mild | | Moderate | | Severe | |
|---|---|---|---|---|---|---|---|
| | | n | % | n | % | n | % |
| Age group (years) | 20–30 | 617 | 87.3 | 90 | 12.7 | 0 | 0.0 |
| | 31–40 | 0 | 0.0 | 486 | 72.9 | 181 | 27.1 |
| | 41–50 | 0 | 0.0 | 103 | 33.9 | 201 | 66.1 |
| Body mass index (kg/m$^2$) | Normal | 436 | 87.0 | 65 | 13.0 | 0 | 0.0 |
| | Over weight | 181 | 19.1 | 479 | 50.5 | 289 | 30.5 |
| | Obesity | 0 | 0.0 | 135 | 59.2 | 93 | 40.8 |
| Year of service (years) | 2 to 5 | 617 | 85.2 | 107 | 14.8 | 0 | 0.0 |
| | 6 to 10 | 0 | 0.0 | 500 | 67.1 | 245 | 32.9 |
| | 11 to 15 | 0 | 0.0 | 72 | 34.4 | 137 | 65.6 |
| Level of physical activity | Sedentary | 0 | 0.0 | 110 | 34.8 | 206 | 65.2 |
| | Light | 0 | 0.0 | 301 | 82.0 | 66 | 18.0 |
| | Moderate | 395 | 51.1 | 268 | 34.7 | 110 | 14.2 |
| | Vigorous | 222 | 100.0 | 0 | 0.0 | 0 | 0.0 |
| Role in the military | Lieutenant colonel | 1 | 100.0 | 0 | 0.0 | 0 | 0.0 |
| | Major | 1 | 33.3 | 1 | 33.3 | 1 | 33.3 |
| | Captain | 0 | 0.0 | 1 | 33.3 | 2 | 66.7 |
| | 1st lieutenant | 1 | 50.0 | 0 | 0.0 | 1 | 50.0 |
| | Master sergeant | 59 | 29.4 | 90 | 44.8 | 52 | 25.9 |
| | First class sergeant | 71 | 38.4 | 77 | 41.6 | 37 | 20.0 |
| | Sergeant | 93 | 40.3 | 100 | 43.3 | 38 | 16.5 |
| | Vice sergeant | 73 | 36.3 | 77 | 38.3 | 51 | 25.4 |
| | Corporal | 146 | 36.9 | 155 | 39.1 | 95 | 24.0 |
| | First class private | 81 | 38.6 | 76 | 36.2 | 53 | 25.2 |
| | New recruit (JUNDI) | 91 | 37.1 | 102 | 41.6 | 52 | 21.2 |
| Smoking | Smoking | 371 | 91.8 | 15 | 3.7 | 18 | 4.5 |
| | Non-smoking | 246 | 19.3 | 664 | 52.1 | 364 | 28.6 |
| Pre-existing conditions | None | 559 | 64.2 | 113 | 13.0 | 199 | 22.8 |
| | Yes | 58 | 7.2 | 566 | 70.1 | 183 | 22.7 |
| Pittsburgh quality of sleep index | Good sleep | 617 | 56.5 | 353 | 32.3 | 123 | 11.3 |
| | Poor sleep | 0 | 0.0 | 326 | 55.7 | 259 | 44.3 |
| WHO- five well-being index | Low risk | 617 | 74.5 | 208 | 25.1 | 3 | 0.4 |
| | Low-medium risk | 0 | 0.0 | 293 | 82.3 | 63 | 17.7 |
| | Medium risk | 0 | 0.0 | 11 | 26.8 | 30 | 73.2 |
| | High-medium risk | 0 | 0.0 | 49 | 35.8 | 88 | 64.2 |
| | High risk | 0 | 0.0 | 118 | 37.3 | 198 | 62.7 |

**Notes.**

*n*, number; %, percentage; WHO, World Health Organization.

Sidiq et al. (2021), *PeerJ*, DOI 10.7717/peerj.12249

**Table 2  Correlation matrix.**

| | | 1 | 2 | 3 | 4 | 5 | 6 | 7 | 8 | 9 | 10 | 11 |
|---|---|---|---|---|---|---|---|---|---|---|---|---|
| 1 | Level of physical activity | | | | | | | | | | | |
| 2 | Role in the military | .007 | | | | | | | | | | |
| 3 | Smoking | −.272** | −.023 | | | | | | | | | |
| 4 | Co-morbidity | −.547** | −.012 | .514** | | | | | | | | |
| 5 | Number of days lost (days) | −.044 | −.036 | .568** | .458** | | | | | | | |
| 6 | Age | .036 | −.022 | .624** | .385** | .838** | | | | | | |
| 7 | Years in service | .033 | −.022 | .612** | .382** | .831** | .984** | | | | | |
| 8 | BMI | −.104** | −.036 | .442** | .357** | .644** | .681** | .688** | | | | |
| 9 | Well being index 5 (WHO) | −.096** | .009 | −.423** | −.175** | −.729** | −.862** | −.864** | −.633** | | | |
| 10 | RMDQ | .044 | −.018 | .520** | .357** | .902** | .834** | .828** | .617** | −.740** | | |
| 11 | Pittsburgh quality of sleep | −.044 | −.048* | .517** | .425** | .867** | .735** | .724** | .566** | −.614** | .790** | |
| 12 | Working hours | .163** | .024 | −.664** | −.624** | −.875** | −.828** | −.818** | −.681** | .693** | −.813** | −.779** |

**Notes.**

BMI,  body mass index; RMDQ,  Rolland Morris Disability Questionnaire.

Spearman's rho correlation is significant at the 0.01 level (2-tailed). $N = 1678$.

*$p < .05$.

**$p < .01$.

study conducted by Chan et al. among Malaysian military personnel ($n = 330$) reported a prevalence of 48.9% of LBP. However, LBP was assessed through a questionnaire (Nordic Musculoskeletal Questionnaire) without identifying the type and region of pain. The study also included female participants ($n = 27$, 8%) and LBP was associated with smoking, history of LBP, history of the accident, lifting weight, and job-related activities (*Chan et al., 2019*). Another study conducted by *Monnier et al. (2015)* among ($n = 272$) Swedish armed force marines reported 19.9% and 36.0% musculoskeletal pain originating from the thoracic and lumbar region of vertebral column, respectively. The present study included 97% of male marine soldiers. A study conducted by Vun et al. in 2018 among navy, army, and air force military personnel ($n = 6,696$) in Canada reported a significant association between the mental health of armed forces and the intensity of back pain. The present study included male ($n = 5,773$) and female ($n = 923$) participants and reported 23% of chronic pain arising from the lumbar region of the spine (*Vun et al., 2018*). A study conducted by Cardoso et al. in 2018 in Brazil reported a back pain prevalence of 58.8% among military police battalion, affecting their daily living activities. However, participants ($n = 97$) with scoliosis, anterior head; lumbar and cervical hyperlordosis; thoracic hyperkinesis, and decreased spinal curvatures were included. The study classified disability due to back pain based on scores obtained from the Oswestry Disability Index (*Cardoso et al., 2018*). A study by *Hou et al. (2013)* reported that the prevalence of LBP among Chinese ($n = 16,42$) soldiers were highest among armored forces (51.3%) than in the artillery (27.5%) or infantry (11.9%). The authors of the present study also reported that the prevalence was higher among participants involved in night training, cross-country race, and grenade-throwing. There was no formal assessment conducted for back pain. Participants in the study were merely asked to recall if they had back pain in the previous two months (*Hou et al., 2013*).

A study conducted by Ernat et al. in 2012 among active infantry soldiers in the USA reporting acute back pain noted that the soldiers seldom seek medical care as they think that the back pain is a normal part of their occupation. The low prevalence of NSCLBP in our study might be due to a similar perception among Saudi male soldiers. However, future research is needed to explore the perception of Saudi military about LBP. The authors of the present study also reported the incidence of LBP about 39.9 per 1000 person-years among all infantrymen. However, the data were extracted from the physician's retrospective medical records of military personnel (*Ernat et al., 2012*).

The percentage of participants who reported smoking in our study was 24.7% ($n = 404$). Smoking was moderately correlated to the number of days lost due to back pain, higher scores on the back pain disability questionnaire, higher BMI, and lower wellbeing index. About 371, 91.8% of active cigarette smokers out of 404 belonged to the age group of 20–30 years and were suffering from mild to moderate back pain disability. These results were supported by a meta-analysis of cross-sectional studies revealing a strong association between smoking and the prevalence of low back pain (*Shiri et al., 2010*). The authors of the present study also reported that incidence of LBP was more substantial in adolescents than in adults. Therefore, an early counseling session to quit smoking at an early age would improve the overall health of younger military recruits (*Pirie et al., 2013*).

Physical fitness is of paramount importance among people undergoing or seeking military training. Military personnel complete strenuous physical training to meet the demanding role in various positions. Our study found a significant variation in the level of physical activity performed at different ranks in the military. We found that the level of physical activity was negatively correlated with the age group. Participants engaging in mild to moderate physical activity reported statistically significant improvement in their quality of life (WHO-5) compared to sedentary level class. It was also noted that participants engaging in a vigorous level of physical activity only experience a mild form of disability due to LBP. The level of physical activity among participants was limited due to the COVID-19 pandemic. It could be the possible reason behind a lack of significance between activity level and amount of back pain disability.

In our study, participants in the age group of 41–50 years reported 66% severe disability due to back pain compared to younger age groups. In contrast, research conducted among US veterans ($n = 67,696$) reported that young male veterans (18–39 years) were 3.1 times more likely to report severe pain than the older age group (*Nahin, 2017*).

We found that poor quality of sleep among participants reporting moderate to severe back pain disability. Similarly, research conducted in US military hospitals among 757 participants reported that sleep disorder is significantly associated with LBP-related health visits among the military population (*Rhon et al., 2019*). In addition, a cohort study conducted among adults reported improvement in LBP with increased sleep quality (*Kovacs et al., 2018*). However, there is a lack of research articles regarding the quantity and quality of sleep required to recover from strenuous military drills.

There were no significant differences in the severity of back pain disability due to the role performed in the military. However, a higher number of corporals reported mild back pain disability when compared to other ranked officers. The sedentary administrative level of activity performed at the military offices by the corporal–ranked officers could be the possible reason behind the high prevalence of NSCLBP.

The quality of life assessed by WHO-5 showed that participants reporting mild back pain disability were at minimal risk, and those with severe back pain disability were at substantial risk of developing poor health-related quality of life.

## LIMITATIONS

There are many limitations to this cross-sectional study. There is a bias involved in self-reported questionnaires. However, validated Arabic-translated questionnaires were used to overcome the language barrier. The type of military-specific activities was not evaluated for each participant. The novel Coronavirus pandemic during the study period may have affected the reported level of activity among participants. The inclusion of only male soldiers in our study limits transferability to other militaries. Moreover, the physiological characteristic such as frequency of previous episodes, intensity of pain and remission period was not assessed among participants. The data were obtained from one military base in Saudi Arabia, therefore limiting the generalization to the entire military population in Saudi Arabia. However, the Hafar batin military city in Saudi Arabia is the largest military base in Saudi Arabia.

## CONCLUSIONS

Our study reports a high prevalence of pain disability originating from the spine (46.8%) among military personnel in Saudi Arabia with a 2.7% prevalence of NSCLBP. Strong associations were found between back pain disability and age group, role in military, smoking, BMI, pre-existing conditions, quality of sleep, and WHO-5 among military personnel. The data can be used to develop risk management strategies such as smoking cessation programs at the military base to reduce the prevalence of NSCLBP. The research data can be utilized to update the health care policy for early rehabilitation of military personnel exhibiting signs and symptoms of NSCLBP. The research data can be used to conduct future research on association between risk factors and NSCLBP among military population globally.

## ACKNOWLEDGEMENTS

We would like to thank the academic and training department of Northern Area Armed Forces Hospital, especially the research unit, for supporting and encourage us through the research process.

### Funding
The authors received no funding for this work.

### Competing Interests
The authors declare there are no competing interests.

### Author Contributions
- Mohammad Sidiq, Mohammad Qasim and Marisia Paz Lopez conceived and designed the experiments, analyzed the data, prepared figures and/or tables, authored or reviewed drafts of the paper, and approved the final draft.
- Wadha Alenazi, Mehrunnisha Ahmad, Suresh Mani, Mohammad Abu Shaphe and Shabir Ahmad Mir conceived and designed the experiments, performed the experiments, analyzed the data, prepared figures and/or tables, authored or reviewed drafts of the paper, and approved the final draft.
- Faizan Z. Kashoo conceived and designed the experiments, analyzed the data, authored or reviewed drafts of the paper, and approved the final draft.
- Omaymah Khodairi and Abdulqader Almutairi conceived and designed the experiments, performed the experiments, analyzed the data, authored or reviewed drafts of the paper, and approved the final draft.

### Human Ethics
The following information was supplied relating to ethical approvals (i.e., approving body and any reference numbers):

The Institutional Review Board of King Fahad Medical City approved the study with IRB log no.19-033E.
## Ethics

The following information was supplied relating to ethical approvals (i.e., approving body and any reference numbers):

The Institutional Review Board of King Fahad Medical City approved the study with IRB log no.19-033E.

## Data Availability

The raw data are available in the Supplementary File.

## Supplemental Information

Supplemental information for this article can be found online at http://dx.doi.org/10.7717/peerj.12249#supplemental-information.

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
