# Peer review of "Prevalence of non-specific chronic low-back pain and risk factors among male soldiers in Saudi Arabia"

_PeerJ, doi:10.7717/peerj.12249_

## Round 0.1 · original submission · Major Revisions

Author(s), both reviewers have recommended "Major Revisions" to your manuscript however I believe you can meet all of their recommended changes.

·

Basic reporting

While the authors should also be commended on the translation of this paper to English, punctuation and grammar do still need to be addressed. The grammatical errors are generally those that come from translating a paper to English and should not be seen as a disqualifier for this paper.

Figures/Tables: There appears to be a numbering sequence issue with the tables that may be a systems error (ie Table 3? Not mentioned in text, and does not exist, but is captioned).

Experimental design

Methods do need some additional detail. If these can be addressed then the paper can progress.

Validity of the findings

This may be improved by improving the grammar and clarity

Additional comments

The study reports on the prevalence of NSLP and associated risk factors in male soldiers in Saudi Arabia.
Overall, the study shows promise with a large sample size and wide representation of military personnel represented. The authors should also be commended on the translation of this paper to English.
GENERAL:
The term ‘incidence’ is used incorrectly. While prevalence is a % of a given population, incidence is time dependent (e.g., per 100 person-years). As such lines 41 and 223 are incorrect as NSLBP incidence cannot be a %.
In the abstract the sample size is listed as exactly 62,000 (also in the discussion – line 159). This sample size is not mentioned in the results, nor how it was determined in the methods. Please correct.
In the methods it states, ‘This cross-sectional correlation study was conducted on military personnel at king Khalid military city hospital, Hafar Al-Baten, Saudi Arabia’ (lines 82-83) but in Figure 1
there is mention of a three military bases from which the sample population is drawn. Please explain this relationship (between the hospital and bases) in the methods and results.
While English readability is established, punctuation and grammar do still need to be addressed. The grammatical errors are generally those that come from translating a paper to English.
Punctuation.
• A full stop may be missing, or in the incorrect position (same with commas): E.g., lines: 71, 74, 76, etc.
Grammar
• Single words used incorrectly or with the incorrect tense: E.g., line 42, ‘associated to’ should be ‘associated with’ and line 51 ‘complain’ should be ‘complaints’; other examples include lines 95, 100-101, 117, 211, etc.
• General sentences varying in tense or singular/plural, etc., : E.g., lines: 88, 90-92, and 133-134.
• When using lists, that is listing several factors, the words ‘and’ or ‘or’ should be applied between the second last and last item on the list. For example: Lines 60-61 (‘…psychological stress, history of back pain, and pre-existing conditions.) and lines 96-98 (‘…such as hypertension, or diabetes….’).
• When using acronyms like LBP and NSLBP use them consistently throughout. For example, while LBP as an acronym is used in line 51 (start of the introduction), the term low back pain is still written though out. For example: Lines 71 (acronym introduced again), 88, 162, 174, 175, etc. The same occurs with NSLBP.
References in text:
• When citing more than one reference, only a single set of brackets should be used. For example, line 53 ‘(Alnaami et al., 2019) ( Cunningham et al., 2006) should be ‘(Alnaami et al., 2019; Cunningham et al., 2006). Other examples are lines 63-64, & 71.
• Spaces missing between word and reference (e.g., RMDQ(Makie et al, 2014)). Other examples are found in lines 107 and 111.

SPECIFIC COMMENTS
Introduction:
The introduction currently reads as one very long paragraph. I suggest separating into several paragraphs. Potential new paragraphs could arise form line 64 (Military service begins…) and Line 74 (Several risk factors….).

Methods:
Lines 83-84. Why only male participants? Please give a reason as this limits transferability to other militaries (should also be noted in the limitations).
Results:
Line 127: The grouping of ‘Obese’ is introduced. How was this group determined? BMI? Please detail in the methods. Also introduce the age bins in the methods.
Lines 138 – 141 & 145-146: Details methods of undertaking the statistical analysis and should be reported in the methods.
Discussion:
Lines 161-170: Provide data from other populations. This data however is prevalence of LBP as opposed to NSLBP. It would be useful to present you clearly state your prevalence of LBP (as opposed to NSLBP only) and clarify if the other studies included the sacral region in their data. There is also little discussion of these studies as opposed to an almost list like delivery. What are the population sizes, did they include female soldiers, were they trainees or qualified officers? Etc
Lines 172-176: Supporting references are needed. Eg. ‘These results are supported by a similar study…’ Which study?
Line 184-186. A recommendation is made regarding early counselling to quit smoking. Is there any evidence to support this intervention is effective?
The limitations should also include lack of a female sample limiting transferability.

Reviewer 2 ·

Basic reporting

The manuscript requires professional editing

Experimental design

The methods section requires sufficient detail which the authors did not provide

Validity of the findings

Validity of the results affected by the poor reporting of the method section

Additional comments

Reviewer Comments

Thank you for the opportunity to review this manuscript. The authors did well in identifying an area of importance in Saudi Arabia, evaluating the prevalence and associated risk factors among male soldiers in Saudi Arabia.

Although the topic is worthy of research and concrete, there is more detail I would like to see in the introduction and methods section; as well as integration of a few additional references and concepts throughout the manuscript.

As such, this manuscript will require some revisions to meet the criteria for publication especially considering that the area of non-specific recurrent low back pain among soldiers is well known and well-researched with plenty of evidence from both developed and developed countries.
Therefore, I would like to read more on the unique contribution of this paper to the existing literature on low back pain among soldiers and there should be a very strong contextual justification presented in the introduction warranting the conduction of the study in Saudi Arabia besides the fact that nothing/litte has been documented about the topic.

As the research design is observational, the authors are recommended to review and apply the STROBE guidelines when writing this manuscript (available via the equator network at http://www.equator-network.org/reporting-guidelines/) to the manuscript.

Please check carefully for grammar/syntax errors throughout the manuscript. Certain sentences are difficult to read in some places. I have illustrated where changes should be made in the main manuscript on my tracked changes document.

Specific comments

Introduction
1. This paper needs to strongly emphasize its unique contribution to the literature base in the context of what other authors have already researched and reported especially studies from developing or developed countries with regards to the prevalence and associated factors to recurrent low back pain among soldiers (Hou et al; Chan et al;)
2. The theoretical perspective to the problem is narrow in light of the many prevalence studies that have been conducted to answer the research question the authors are posing. There is plenty of evidence on the prevalence of low back among soldiers from developing countries (information which can be applied to Saudi Arabia). It has to be made clear why such data are not applicable to Saudi Arabian soldiers. Novelty and originality concerns will be answered if such concerns are tackled.
3. The authors should also highlight the contextual background underpinning the conduction of the study in Saudi Arabia. What prompted the authors to want to do this study in Saudi Arabia? What local problem identified or observed which justifies this study?
4. The authors are encouraged to put some statements on how studying the prevalence and the preselected associated factors of NSLBP would significantly assist with policy, clinical practice, medical, rehabilitation management, and prevention of NSLBP, etc in Saudi Arabia. In other words, provide a hint of the significance of this study in the Saudi Arabia context and the implications of currently not knowing such data in that country

Methods
1. The method section should provide substantially more detail about your research setting, data collection procedures. Ideally, there should be a clear and logical step-by-step description of your study procedures.
2. Put your research setting in context for other authors from the rest of the world to appreciate the setting. Why was the setting used????
3. Were all participants are screened for inclusion criteria????
4. How many participants were invited to participate in the study through pamphlets, email, and advertisements across the offices and residential areas.
5.Justify the selection of LBP definition used in the study? Why at least 12 weeks??? What about pain intensity???
6. "Participants who were currently suffering from non-specific low back pain were eligible to participate in the study" Please justify why "currently suffering"
7.Who did the physical examinations helping with LBP diagnosis? What were they examining? Be very specific
8. Attach the English version of the questionnaire
9.Justify the selection of the tool for assessment of QoL
10. Please comment on the following
Sample size considerations??
Target population
Sampling strategy
Justification of the instruments [psychometric properties of the tools]
Procedure missing
Statistical analysis plan missing

Results

Consider putting the Response rate
Consider the use of the flow diagram
Provide sociodemographic details of the participants/baseline information of the participants
Define this region [pain originating from the lumbosacral region] in relation LBP region??? how different is it from the low back region
Very low prevalence of NSLBP. This questions the contextual justification for the study
Justify the use of Spearman's rank-order correlation

---

## Round 0.2 · Minor Revisions

Dear Dr Sidiq and Dr Kashoo, your revised manuscript has undergone a thorough review and the Reviewers have further issues which must be addressed. Please see their comments and recommendations. Thanks, A/Prof Mike Climstein

·

Basic reporting

Grammar is much improved but does need some final work. No other concerns

Experimental design

No concerns

Validity of the findings

There are also a few areas that need clarification when being compared to wider literature that is not clear and may be misrepresenting the results:

Lines 233-236: This is not clear. How is the NSCLBP of your study supported by Ernat et al., (2012) whose study was on LBP? There were even notable differences in numbers reporting LBP and NSCLBP in this study – 46.3% versus 2.7% so how your study related to that of Ernat et al., (2012) is not clear.

Lines 260-261: This is not clear. How does your study relate to Nahin (2017) given that you are reporting NSCLBP as opposed to LBP as per their study (noting above limitations). Furthermore, the age group in your study 31-40, which also falls into the bracket by Nahin, (2017) of 18-70, was 27%? So I am unsure why the comparison of your age group 41-50 (66%) is more valid to compare to 18-70 year olds (Nahin, 2017) and be considered similar while the 31-40 age group (27%) is not?

Additional comments

Thank you for the opportunity to again review the study titled: Prevalence of non-specific chronic low-back pain and associated risk factors among male soldiers in Saudi Arabia. The paper is much improved.

General Comments:
Some general errors and inconsistencies can be addressed throughout the paper:
• Terminology: Please select either ‘subjects’ or ‘participants’ (see Line 110 for an example where both are used)
• There are inconsistencies in Capitalisation (e.g. WHO five Well-being - Line 130; Five Well-Being Index (WHO-5) - Line 145 and Who five Well-being Index - Line 149).
• Change all ‘Date was..’ to ‘Data were…’ (Data is plural, datum is singular) – See examples, Line 156, 237, 284, etc
• Be consisted with acronyms eg., Line 84 – NSLBP, Line 168 – NSCLBP, Line 170 – CNSLBP
• Please state ‘who’ for sentences starting with ‘ A study conducted…’ E.g.
• Please add a citation after sentences that start with terms like ‘The study… ‘ E.g. Line 213, 218, 222, 226, 233…etc
• When using the term ‘they’ please clarify who ‘they’ are. E.g., Line 229, 236, 245, etc

Specific Comments:
ABSTRACT
Line 33: ‘…referral from the other department…’ Please clarify what ‘other department?’ and ‘other’ in relation to which original department?
Line 39: Recommend rewording to: ‘All eligible participants were assessed for demographic variables and associated risk factors and complete the Rolland Morris Disability Questionnaire and WHO-Five Well-Being Index.’
NOTE: There is no need to introduce the (RMDQ) acronym here as it is not used again in the ABSTRACT
Line 41: Recommend rewording to: ‘Results: This study identified a 46.3% prevalence of pain…’
Lines 43-45: My thoughts only: It may increase flow to reorder the factors associated with NSCLBP but strength of relationship. It allows the reader to easily identify those with the strongest and weakest relationships.
Line 46: Please add the word ‘male’ between ‘among’ and ‘Saudi’

INTRODUCTION
Lines 61 and 63: Please replace ‘are’ with ‘include’ (e.g., ‘Individual factors include the sedentary….’)
Line 79: Please add the word ‘strategies’ after ‘prevention’
Line 81: Recommend rewording to: ‘Conversely, untreated NSCLBP might lead to…’
Line 84: Recommend rewording to: ‘…treating NSCLBP as soon as possible to onset would significantly…’
Line 85: Remove sentence relating to no research conducted in ME. This is made clear and is well positioned Lines 88-89
Line 90-91: Remove sentence relating to prevalence studies as it distracts from the flow (i.e., no research so this is the aim of our research). If you think it is important to include perhaps place it at the start of Line 86.

MATERIALS AND METHODS
Line 100: Recommend rewording to: ‘However, a military base in Jeddah province, Saudi Arabia, are recruiting female soldiers, but that research will not form part of this study.’
There is no mention of the ‘referral from the other department’ (Line 33) in the methods.
Line 105: Recommend rewording to: ‘…soldiers reporting with LBP…’
Line 107: Recommend rewording to: ‘…which lasted for at least…’
Line 108: Recommend removing ‘…and is reported to affect health and wellbeing…’ as it is not relevant to your classification and is a distraction
Line 111-112: Correct to either ‘Physical Therapist’ or ‘physical therapist’
Line 115: Recommend rewording to: ‘…radiographic imaging…’
Line 117: Recommend rewording to: ‘…with a prolapsed inter-vertebral…’
Lines 124-127: Recommend rewording to: ‘…demographic data such as age, weight, height, years of service and interview for role in the military and history of smoking. Medical records were..…’
Line 128: Correct reference (World Health Organization, 2000)
Line 136: ‘…with the English version…’
Line 147: Recommend rewording to: ‘…The score was multiplied…’
Line 151: Remove ‘The’ before ‘Physical activity….’

RESULTS:
Line 165 and 167. – Remove Line 165 and include it into Line 167..e.g., ‘Out of 62,000 personnel contacted for the survey, 8,706 (46.3%) reported with LBP and of these 1,687 (2.7%) reported with NSCLBP…’
Line 147: Recommend rewording to: ‘… (22.8%) or severe intensity NSCLBP…’
Line 174: Recommend rewording to ‘…as compared to the older age group…’
Line 179: Add ‘and’ after ‘(n=216, 54.5%)’ and ‘engaged’
Line 180: Remove n= (typically only used in brackets)
Line 188: Remove extra punctuation
Lines 202-205: Sentence does not make sense – please reword
Lines 233-236.: This is not clear. How is the NSCLBP of your study supported by Ernat et al., (2012) whose study was on LBP? There were even notable differences in numbers reporting LBP and NSCLBP in this study – 46.3% versus 2.7% so how your study related to that of Ernat et al., (2012) is not clear.
Lines 260-261: This is not clear. How does your study relate to Nahin (2017) given that you are reporting NSCLBP as opposed to LBP as per their study (noting above limitations). Furthermore, the age group in your study 31-40, which also falls into the bracket by Nahin, (2017) of 18-70, was 27%? So I am unsure why the comparison of your age group 41-50 (66%) is more valid to compare to 18-70 year olds (Nahin, 2017) and be considered similar while the 31-40 age group (27%) is not?
Line 272-273: Please clarify who the ‘they’ are and if you mean ‘…The high number of Corporal-ranked officers reporting LBP could be because other ranked officers were engaged in sedentary…?
Lines 275-277: Risk of what. Please expand
Lines 282: Please change ‘corona’ to ‘Novel Coronavirus’ as defined by the WHO and is listed in the ICD.

CONCLUSIONS:
Consideration: I think it would be worthwhile to include the use of this data to inform risk management strategies, like the smoking reduction one mentioned earlier in your article (Lines 247-248).

REFERENCES:
Please correct reference LINE 396 Should be World Health Organization.

Reviewer 2 ·

Basic reporting

There are few instances in need of professional editing. See attached manuscript for areas in need of corrections. Check your grammar, use of comma and spellings, and use of correct abbreviations.

So many sentences requiring referencing in the manuscript. Kindly see the attached document. For example, the following sentences in the background section require referencing at the end.
"More than half of the general population seeks medical advice for LBP at some point in their lives"
"Back pain without any known cause for 12-weeks is termed non-specific chronic low back pain (NSCLBP)"
"The medical cost involved in the management of NSCLBP is relatively lower than the more severe form of LBP".
"Therefore, identifying and treating NSLBP at the earliest would significantly reduce the medical cost and absenteeism from work"

The authors should attach the English low back pain questionnaire as a supplementary or additional file and evidence of pilot study and pre-testing of the instrument should also be mentioned and highlighted in the manuscript. This is in light of the very low prevalence rate of NSCLBP reported in the study. Instrument development was not mentioned in the study methods section as well.

The main issue with the background as highlighted in my first review is the lack of a comprehensive theoretical and contextual background underpinning the conduction of the study. Following the author's corrections, still, that aspect is not very clear from the background. For example, the authors mentioned that "Several risk factors have been reported to be associated with NSCLBP among soldiers" and they went on to give us the risk factors based on the results from a study by Simula et al., 2019. So if we know this, why are repeating this study, investigating again the associated factors? What makes the soldiers in Saudi Arabia unique from the rest of the soldiers from other countries like Sweden, China, USA, Canada, Brazil??? Why can’t we extrapolate from the findings from these previous studies and apply them in Saudi Arabia? In addition, given the fact we know already that soldiers are predisposed to low back pain by nature of their military training (Chan et al., 2019; (Monnier et al., 2015; Vun et al., 2018; Cardoso et al., 2018; Hou et al., 2013; Ernat et al., 2012), what contextual evidence exists in Saudi Arabia especially at the military base the study was conducted on that justifies the need for this study. This tackles originality and novelty issues and advances new information rather than plain replication of what other authors have already established previously and just changing the setting. Meaningful replication is encouraged where the authors uniquely specify the contextual problem, how it was identified, local consequences of the problem unabated and the significance of addressing that local problem.

The authors are strongly reminded that "To the best of our knowledge, no research is published about NSCLBP and its associated risk factors in the Saudi military population" is not a strong justification or problem statement warranting the conduction of research.
They ought to highlight to the global audience, what they observed as a problem in Saudi soldiers, the impact of the problem, and how this study will address some of the challenges posed by the problem.

Experimental design

"Only male soldiers were enrolled in this study using a convenient sampling method" Based on this, justify why only male soldiers were enrolled and with such a huge sampling frame wy convenience sampling was used? Discuss the limitations of that sampling strategy as well as the generalisability of the results.



Be very clear whether your n=62 000 reflect the number of male soldiers only at Hafar Al-Baten, or represents the entire population of soldiers at the military base.

Comment about the inter-rater reliability of the 4 PTs who conducted the physical examinations. In addition, it's not clear what they actually did as part of the physical examination. This sentence is confusing given the fact that they examined for LBP ". The physical examination included motor, sensory, and reflex examination of the upper limb and lower limb"

"The primary outcome of the study was the prevalence of NSCLBP among military personnel in Saudi Arabia. Assuming a 95% confidence level, with an expected prevalence between 53.2% to 79.17% LBP based on an earlier study (Awaji, 2016) with 0.05 absolute precision, the required sample size was between 255 to 382 military personnel. However, the questionnaire was distributed to the military population at the military base comprising 620000 military personnel"
Why distribute the questionnaire to 62 0000 when you only needed a maximum of 382 soldiers??

Include a statement on how the low back pain questionnaire was developed, translated, and validated

Attach the low back pain questionnaire for the readers to see how the screening question for NSCLBP was asked. What was the specific question screening for chronic low back pain asked in the questionnaire?

Write RMDQ in full first time

"The Arabic version of PSQI demonstrated good reliability, (Cronbach α = 0.77) acceptable internal consistency, (Cronbach α = 0.65) and moderate to a high correlation between global PSQI and five components of PSQI scores" Check the correctness of this sentence

"The physical activity level was categorized into four levels based on participation in military drills" Categorized by who?????????

"The ordinal scale data were analyzed using Spearman rank-order analysis" This sentence is incomplete and incomprehensible

"A total of n= 8706 responded with back pain with a 46.3% response rate" Please check this sentence in relation to what you wrote under results

"Out of 1678 participants, 617 (74.5%) had mild; 679 (40.5%) moderate; 382 (22.8%) sever intensity of the CNSLBP" Please check your spellings here and consistent use of abbreviations here

Was there an association between age and pain severity???

What about the relationship between age and prevalence of NSCLBP?? Was the prevalence increasing with the increasing age category statistically speaking??

Was there an association between BMI and pain severity??????

Was there an association between level of physical activity and pain severity???

Validity of the findings

"The low prevalence of NSCLBP in this study could be due to stringent inclusion criteria". Explain how the stringent inclusion criteria accounted for the low prevalence of NSCLBP?

Avoid repeating results in the discussion section.

Additional comments

Thank you for the opportunity to re-review the manuscript.

The manuscript still requires some significant revisions on the background, methods, and discussion section.

I have attached a document with some track changes with more comments

Annotated reviews are not available for download in order to protect the identity of reviewers who chose to remain anonymous.

---

## Round 0.3 · Minor Revisions

Thank you for addressing the revisions, only minor further revisions and a comment are required (some made in the attached manuscript via track changes). Thanks, A/Prof Mike Climstein

---

## Round 0.4 · Minor Revisions

The Reviewers have noted only "minor" revisions requred by the authors. Please address all of these issues and resubmit your amended manuscript. Thanks, A/Prof Mike Climstein

·

Basic reporting

No concerns

Experimental design

No concerns

Validity of the findings

No concerns

Additional comments

The paper is much improved however there are still some minor areas that need addressing, mostly grammatical in nature

General Feedback:
• Check punctuation. Often punctuation errors are hard to identify with all the track changes. By selecting ‘no markup’ in the review menu this can be rectified. Examples include: Lines 50, 124, 139, 210, 235, 248, 311.
• The WHO outcome measure is still presented inconsistently throughout. Please select only one term for the outcome measure. Examples include: Line 41 versus 156 versus 184 versus 302 versus 318.
• Remove all instances of ‘n=’ in text unless within brackets (e.g. Line 203 should be ‘Out of 404 active smokers’ while line 205 is correct with ‘…such as diabetes (n=141,8.4%)).


Specific Feedback
Line 33. The use of ‘other’ is confusing. Suggest rephrasing to ‘ …from in-patient and out-patient departments of the military hospital…’
Line 43: Add ‘s’ after correlation as you are conducting more than one (e.g., with age, quality of sleep, etc).
Line 75-76. Grammar needs correction. Suggest ‘ …due to their high physical demands of their profession. Active military personnel usually carry heavy equipment during training which can lead to over-stressing of their musculoskeletal system, especially the spine.’
Line 78: Add ‘the’. ‘…among the general public…’
Line 81: Please rephrase, I do not understand this sentence.
Line 93: The Acronym SA is introduced without preamble (i.e., Saudi Arabia (SA)) and then not consistently used (e.g. Line 96). I suggest removing all SA use.
Line 97: Suggest rephrasing to ‘…73.9% of health care workers reported suffering from LBP…’
Line 100: Grammar and wording. Suggest correcting and rephrasing to ‘Military personnel are at a greater risk of developing LBP due to…’
Line 102: Clarity. Please remove the term ‘militia’ as this is different to ‘military’. A militia is typically an Army of trained civilians.
Line 109. The term ‘analytical’ can be removed as most research studies include analysis and this is made clear in your methods.
Line 50: Grammar: Correct to ‘interviewed’
Line 151: Grammar: Remove ‘The’ before BMI.
Line 171: Grammar: Remove ‘The’ before ‘Quality of life…’
Line 172: Grammar: Change ‘the’ to ‘their’

Line 174: Grammar: Change ‘100 stand’ to ‘100 stands’
Line 191: Grammar add ‘the’ before ‘spine’.
Line 193: Move the standard deviation before the SI unit ‘years’
Line 194: Grammar: Consider changing to ‘ …had mild, 679 (40.5%) moderate, or 382 (22.8%) severe intensities of NSCLBP.’
Line 201: Remove ‘the’ before ‘Further analysis’
Line 205: Grammar. Add ‘a’ before ‘co-morbidity’.
Lines 205-210: Add commas after list items (e.g., diabetes (n=141, 8.4%), hypertension…’
Lines 208-210: Please rephrase. I do not understand this sentence.
Line 216: Grammar: ‘correlations’
Lines 217-218. Grammar: Please add ‘An’ before ‘ordinal regression…’ and add a ‘s’ after ‘relationship’ and ‘variable’.
Line 226. Grammar: Remove ‘are’ before less.
Line 235: Grammar: ‘reported a prevalence of’
Line 240: Remove ‘(‘ before Monnier et al.
Lines 243, 247, 252: Please add the year after the authors (e.g., Vun et al., (2018)…)
Line 243: Please change ‘armed’ to ‘Army’: An armed force and an Army are two different populations.
Line 258-259: Please consider rephrasing to ‘ in the USA reporting acute back pain noted that the soldiers seldom…’
Line 273: Grammar: Consider ‘…health of younger military recruits…’
Line 280: Grammar: Consider ‘…significant improvements in their quality of life…’
Line 284: Suggest rephrasing to ‘…a lack of significance…’
Lines 290-291: Please rephrase. Sentence is not very clear.
Lines 298-299: Consider rephrasing to ‘…a higher number of Corporals reported mild back pain disability when compared to other ranked officers.’
Line 310: Please add ‘s’ to ‘limit’.
Lines 317: Grammar. Consider rephrasing to ‘Strong associations were found between back pain disability and age group, role in military,…’
Line 468: Swain reference. Please confirm that (xxxx) is correct.

Reviewer 2 ·

Basic reporting

Greater improvement in basic reporting. No further comments on this.

Experimental design

Line 115 should be deleted which reads "However, a military base in Jeddah province, Saudi Arabia, are recruiting female soldiers, however that research will not form part of this study" This statement is not adding value.

Sub-heading "study design" should be "study design, participants and research setting"

The primary outcome of the study was the prevalence of NSCLBP among military personnel in Saudi Arabia BUT the screening question for the identification of the participants with NSCLBP was not very clear from the low back pain study questionnaire. Please clarify. The Nordic questionnaire attached cannot be used to identify patients with chronic NSLBP for 12 weeks since it elicited LBP information about the last 12 months and 7 days.

As a limitation, your adopted definition of NSCLBP only considered "duration" as the only physiological characteristic or parameter and ignored important pain elements such as frequency of episodes, intensity, severity, and remission period. See definition by Stanton et al., 2010.

Validity of the findings

"Soldiers reporting with LBP were screened for inclusion criteria. Inclusion criteria included pain or discomfort originating from the lumbosacral region without any known cause, which lasted for at least 12 weeks, without any radiating or specific spinal diseases. The persistence of NSCLBP for 12 weeks is considered chronic (Hallegraeff et al., 2012).. Participants who were currently suffering from NSCLBP were eligible to participate in the study"


Based on the sentences above, the authors need to clarify whether they only included participants "who had an episode of NSCLBP" at the time of the study and excluded all the others without. How then do you assess the associated factors if you exclude participants without NSCLBP?

Additional comments

None

---

## Round 0.5 · accepted · Accept

Dr. Sidiq and Dr. Kashoo, thank you for making the minor amendments requested by the Reviewers. I am pleased to recommend your manuscript for publication in PeerJ. Thank you for supporting PeerJ and we look forward to further submissions from you and your colleagues. A/Prof Mike Climstein